# The Utility of Ambulance Dispatch Call Syndromic Surveillance for Detecting and Assessing the Health Impact of Extreme Weather Events in England

**DOI:** 10.3390/ijerph19073876

**Published:** 2022-03-24

**Authors:** Simon Packer, Paul Loveridge, Ana Soriano, Roger Morbey, Dan Todkill, Ross Thompson, Tracy Rayment-Bishop, Cathryn James, Hilary Pillin, Gillian Smith, Alex J. Elliot

**Affiliations:** 1South West Field Service, UK Health Security Agency, Bristol BS1 6EH, UK; 2Field Epidemiology Training Programme, UK Health Security Agency, London NW9 5EQ, UK; 3Real-Time Syndromic Surveillance Team, Field Service, UK Health Security Agency, Birmingham B2 4BH, UK; paul.loveridge@phe.gov.uk (P.L.); ana.soriano@phe.gov.uk (A.S.); roger.morbey@phe.gov.uk (R.M.); dan.todkill@phe.gov.uk (D.T.); gillian.smith@phe.gov.uk (G.S.); alex.elliot@phe.gov.uk (A.J.E.); 4Communicable Disease Control Evidence and Epidemiology, Warwick Medical School, The University of Warwick, Coventry CV4 7HL, UK; 5Extreme Events and Health Protection Team, UK Health Security Agency, London SE1 8UG, UK; ross.thompson@phe.gov.uk; 6West Midlands Ambulance Service University NHS Foundation Trust, Brierley Hill DY5 1LX, West Midlands, UK; tracy.rayment-bishop@wmas.nhs.uk; 7Association of Ambulance Chief Executives, London EC4A 4AB, UK; cathryn.james@aace.org.uk (C.J.); hilary.pillin@aace.org.uk (H.P.)

**Keywords:** weather, emergency medical dispatch, hot temperature, cold temperature, climate change, ambulance, public health, surveillance

## Abstract

Extreme weather events present significant global threats to health. The National Ambulance Syndromic Surveillance System collects data on 18 syndromes through chief presenting complaint (CPC) codes. We aimed to determine the utility of ambulance data to monitor extreme temperature events for action. Daily total calls were observed between 01/01/2018–30/04/2019. Median daily ’Heat/Cold’ CPC calls during “known extreme temperature” (identified *a priori*), “extreme temperature”; (within 5th or 95th temperature percentiles for central England) and meteorological alert periods were compared to all other days using Wilcoxon signed-rank test. During the study period, 12,585,084 calls were recorded. In 2018, median daily “Heat/Cold” calls were higher during periods of known extreme temperature: heatwave (16/day, 736 total) and extreme cold weather events (28/day, 339 total) compared to all other days in 2018 (6/day, 1672 total). Median daily “Heat/Cold” calls during extreme temperature periods (16/day) were significantly higher than non-extreme temperature periods (5/day, *p* < 0.001). Ambulance data can be used to identify adverse impacts during periods of extreme temperature. Ambulance data are a low resource, rapid and flexible option providing real-time data on a range of indicators. We recommend ambulance data are used for the surveillance of presentations to healthcare related to extreme temperature events.

## 1. Introduction

Extreme weather events, such as sudden or gradual increases or decreases in temperature, snow, ice, and flooding, have a serious impact on health worldwide [1]. Global climate change is predicted to increase the frequency and intensity of extreme weather, and areas previously less affected will be put at greater risk [1,2] Extreme weather is increasingly recognized as an emerging health threat in the UK, with climate change models showing an increase in the likelihood of heatwave events in the future [3].

Globally, the impact of heatwaves on health has been well documented: 514 heat-related deaths and 3300 excess emergency admissions following the 1995 heatwave in Chicago; 3000 deaths associated with the 1994 heatwave in South Korea; and 71,000 deaths as a result of a European heat wave in 2003 [4,5,6,7]. Similar increases in mortality have been found during extreme cold weather events. A study in China found a 28% increase in the cumulative excess risk associated with non-accidental mortality on cold compared to non-cold-spell days [8]. Extreme weather events have a disproportionate effect on the vulnerable within society and persons with pre-existing health conditions; the very old and young are particularly at risk [1,7].

The heatwave and cold weather plans for England outline surveillance as a key part of the response to extreme weather events [9,10]. The UK and other nations need a diverse range of health data sources and surveillance systems to measure the impact and provide early warning of adverse health outcomes associated with extreme weather events. Syndromic surveillance has been shown to be able to monitor a wide range of health events and provide intelligence on infectious and environmental threats [11,12,13].

Ambulance dispatch data has the potential to provide near real-time and low-cost surveillance of infectious and non-communicable disease [14,15]. Ambulance services provide a first point of contact or access point to persons seeking urgent medical care [14]. We aimed to determine the utility of the National Ambulance Syndromic Surveillance System (NASS), established in England during 2018 [16] to monitor the health impact, provide situational awareness, and provide alerts for when we might observe health effects as a result of extreme and known temperature events in England.

## 2. Materials and Methods

The NASS collects anonymized daily data from all ten ambulance services in England. Services passively submit information to the UK Health Security Agency (UKHSA) which is used daily to monitor 18 syndromes. Syndromes are based on chief presenting complaint (CPC) codes, which are generated in the trust electronic patient management system during an ambulance call. Chief presenting complaint codes are high level triage codes identifying the main clinical problem from information given during an emergency call. The codes enable prioritization of calls so that ambulance services can be dispatched and respond appropriately. The ‘Heat/cold exposure’ CPC calls were identified as an indicator for assessing the impact and health burden of extreme weather events. The ‘heat/cold exposure’ CPC call code related to calls regarding heat or cold exposure problems including hypothermia, heat exhaustion, heat stroke and frostbite; however, the aggregated nature of the CPC means heat and cold problems cannot be disentangled.

Daily ‘heat/cold exposure’ CPC calls were extracted from NASS for the period 01/01/2018–30/04/2019. More recent data were not considered due to the significant impact of COVID-19 on ambulance syndromic calls [17]. Data on daily mean, min, and max central England temperatures, ‘known temperature events’ (KTE; identified *a priori*), Heat-Health and Cold Weather Alerts, and key syndromic indicators from other UKHSA systems (telehealth (NHS 111)), emergency department (EDSSS), GP in hours (GPIH), and GP out of hours’ (GPOOH)) were obtained and linked to NASS call data by date of report [9,10].

‘Extreme temperature days’ (ETD) were defined as those with a temperature outside the 5th or 95th percentiles for the total study period. KTE identified *a priori* were as follows: two periods of severe cold weather (“beast from the east” 24/02/2018 to 04/03/2018 and “mini beast from the east” 17/03/2018 to 18/03/2018) and a heatwave during 2018 (22/06/2018 to 07/08/2018). UKHSA ‘Heat Health and Cold Weather Alert’ data were only calculated from 1 November to 31 March (cold weather alert period) and 1 June to 15 September (heat health alert period) each year and analysis was restricted accordingly [9,10]. There are three levels of alert: level 1, ‘summer/winter preparedness’; level 2, hot or cold weather above normal thresholds is forecast; and level 3, hot or cold weather above normal thresholds is currently being experienced. The levels do not indicate severity of conditions and are not based solely on temperature levels. NASS statistical alarms were calculated using a multi-level hierarchical mixed effect model (RAMMIE). RAMMIE is used daily to produce daily alarms across all English syndromic surveillance systems. In this instance it was used to calculate expected baselines and statistical alarms at the national level [18].

Data were analyzed at England level with daily call data analyzed using median, mean, interquartile range (IQR), minimum, and maximum. Daily heat/cold exposure CPC calls and seven day moving averages were plotted and visually compared to daily temperature data and UKHSA Heat Health and Cold Weather Alerts. The number of heat/cold exposure CPC calls were compared between ETD, KTE, and UKHSA Heat Health and Cold Weather Alerts level (level 1 to 3) days using notched box and whisker plots and Wilcoxon signed-rank test. Regional UKHSA Heat Health and Cold Weather Alerts levels were aggregated to the national level. Aggregation used a hierarchy where the presence of any regional level 3 alert defined that day as “level 3”, followed by any level 2 alerts. For a day to be classified as level 1 (normal reporting) all regions needed to be classified as level 1 on that day. The concordance between NASS alarms generated by RAMMIE (as described above) and UKHSA Heat Health and Cold Weather Alerts was determined by calculating the inter-rater reliability and Cohen’s kappa with 95% confidence interval.

The corrplot and Hmisc packages were used to examine the correlation between the daily NASS call count and corresponding ‘heatwave’ indicator counts from other UKHSA syndromic surveillance systems (NHS 111 ‘heat and sunstroke’, NHS 111 ‘insect bites’, EDSSS ‘heat/sunstroke’, GPOOH ‘sunstroke’, GPIH ‘insect bite’ and GPIH ‘heat stroke’) were calculated using Spearman rank correlation coefficients, mid ranks were used for ties and *p*-values approximated using F distributions. A correlation matrix was produced to show the relationship between indicators [19,20]. As all comparable syndromic indicators were related to heat exposure the analysis was restricted to the period when the Heat Health alerts are in operation in England (1 June to 15 September). All analyses were undertaken using R version 4.0.2 (2020-06-22) “Taking Off Again” [21].

## 3. Results

### 3.1. System Description

The study period consisted of 484 days from 01/01/2018 to 30/04/2019. Data were available from ten ambulance trusts that mapped to nine UKHSA regions in England. During the study period, NASS received 12,585,084 syndromic calls in total. When looking at heat/cold exposure CPC calls, there was a median of 5 in England per day (interquartile range: 3 to 8) and ranging from 0 to 54 calls.

For the majority of the study period, calls relating to heat/cold exposure CPC calls stayed between 0–10 calls per day. However, spikes in the number of calls were observed, particularly during sudden drops or prolonged increases in temperature resulting in corresponding spikes in the number of heat/cold exposure CPC calls (Figure 1). There was a heightened number of calls during the three KTE, two cold weather and one heat wave, with sudden spike in call numbers during the event followed by a drop when the heat wave or cold weather subsided. Furthermore, spikes in heat/cold exposure CPC calls were frequently concordant with the presence of UKHSA Heat Health and Cold Weather Alerts (Figure 1). A substantial spike in heat/cold exposure CPC calls was observed outside a UKHSA Heat Health and Cold Weather alerting period and was not associated with a KTE, however, it corresponded with an increase in temperature (Figure 1).

### 3.2. NASS Heat and Cold Exposure CPC Relationship with Extreme Temperature

There was a median of 16 (IQR: 11 to 28) heat/cold exposure CPC calls per day during ETD (days within 5th and 95th temperature percentile) compared to 5 calls/day (IQR: 3 to 7) during normal temperature days, *p* < 0.001 (Figure 2). While the median was greater in ETD compared to normal days, we observed several days where a high number of calls (max = 42) were received even though temperatures were within the ‘normal’ range. We observed the same significant elevation in the number of daily heat/cold exposure CPC calls during the three KTE (indicated in the grey boxes marked on Figure 1). The large peak observed on the 7 May 2018 relates to the May Bank Holiday.

For analysis of UKHSA Heat Health and Cold Weather Alerts, data were restricted to the 348 days in the study period where these were issued (alerts issued between 1 November to 31 March and 1 June to 15 September). There was a significantly higher median daily number of heat/cold exposure CPC calls on days when a level 2 (median 10, IQR: 6 to 16 daily calls) or level 3 (median 12, IQR: 8 to 22 daily calls) alert was issued compared to level 1 (median 4, IQR: 3 to 6) days (level 1 compared to level 2 or 3 both *p* < 0.001; Figure 2).

The concordance between the presence of a NASS system RAMMIE alarm and a UKHSA Heat Health and Cold Weather level 2 and 3 alert in any UKHSA region had an inter-rater agreement of 82% and Cohen’s kappa 0.47 (95% CI: 0.37 to 0.57), indicating moderate agreement between the two systems. On the majority (67%) of days where a NASS statistical alarm was identified there was a corresponding UKHSA Heat Health and Cold Weather level 3 alert. However, when looking at the 10% of the days when there was no NASS statistical alarm, there was a UKHSA Heat Health and Cold Weather level 3 or 2 alert, respectively (Table 1).

### 3.3. NASS Heat and Cold Exposure CPC Relationship with Other Syndromic Surveillance Systems

Comparable extreme weather syndromic indicators from other UKHSA systems were all heat-related and, therefore, data were restricted to 1 June to 15 September (Heat Health watch period). We found a significant positive correlation between the number of NASS heat cold exposure calls and NHS 111 heat/sunstroke (r = 0.78), NHS 111 insect bites (r = 0.61), EDSSS heat stroke (r = 0.76), GPOOH sunstroke (r = 0.68), and GPIH heat stroke (r = 0.34). NASS heat and cold calls were not significantly correlated with GPIH insect bites (r = 0.19; Figure 3).

## 4. Discussion

We aimed to determine the utility of NASS to monitor the health impact of extreme weather events and assess its relationship with related syndromic indicators. The study found that ambulance heat and cold exposure calls had a strong association with severe weather and temperature events occurring in England. The number of daily calls were generally low, but during extreme weather events heat and cold ambulance calls increased exponentially. The analysis shows that, while the heat/cold exposure CPC calls code is infrequently used by ambulance services, it is highly specific and sensitive to days where extreme temperature events were observed. The NASS indicator can be used in real time to quantify and describe serious community onset health effects resulting from extreme weather events. The finding is supported by the significant elevation of NASS heat/cold exposure CPC calls during ETD and on days where a KTE had occurred. NASS heat/cold CPC calls were also significantly elevated during UKHSA Heat Health and Cold Weather Alerts level 2 or 3 alerts. UKHSA Heat Health and Cold Weather Alerts are based on meteorological factors such as forecast temperature and wider contextual factors and are considered together as part of a dynamic risk assessment [9,10]. The concordance between NASS statistical alarms and meteorological alerts shows that the system can probably detect the health impact from multi-modal weather and not just temperature events. The added benefit of NASS is that it provided alerts throughout the whole year, whereas UKHSA Heat Health and Cold Weather Alerts were provided only at specific times of year. The benefit of annual coverall of NASS data is particularly apparent when looking at 7 May, 2018, where a significant spike in calls was observed in relation to the hottest ever recorded May Bank Holiday. The temperature increase was not within the UKHSA Heat Health and Cold Weather alert period; however, ambulance attendances potentially indicate a significant health impact during this weekend.

We found that NASS heat/cold exposure CPC calls showed strong correlation with heat-related syndromic indicators in other UKHSA syndromic surveillance systems. Strongest correlations were found with NHS 111, GPOOH, and EDSSS surveillance data. These findings are expected, as these services provide similar ‘unscheduled’ care, which are more likely to be used by patients in the event of severe heat exposure symptoms [22]. The correlation supports the reliability of NASS system to identify health events related to extreme weather. The evidence presented demonstrates that the NASS heat/cold exposure CPC calls are broad and a ‘catch all’ indicator, they are flexible and responsive, and can be used to trigger action. Actions resulting from NASS system alerts could include targeted health messaging to members of the public and alerting emergency services to increase awareness of the health impact of hot/cold weather. The impact of extreme temperatures on health can vary between and within seasons [22]. In order for appropriate public health action to be taken, intelligence is needed, and NASS provides this in near real time.

Gaps in information have a direct impact on our ability to implement comprehensive and effective public health interventions in response to extreme weather events. Syndromic surveillance systems have been shown to identify potential public health threats and direct the action to mitigate [11,23,24,25]. There have been several ambulance dispatch systems described in the published and grey literature [14]. The majority of current systems report at the regional level, utilize retrospective data analysis, and focus on respiratory infections or harms related to drugs or mental health [14,15,26]. Although syndromic data has been used to monitor the health impact of extreme weather, to our knowledge NASS is the only ambulance-based system that can provide intelligence at a national level and a ‘multi-hazard’ response [27]. Its national prospective data collection ensures timely data to monitor the effects of extreme weather across the whole of the country and supports the identification and management of incidents which might be more severe at certain geographies. NASS provides an added benefit as it captures data associated with the severe end of the health spectrum. These events are potentially not recorded in other systems and, therefore, NASS can provide novel real-time data to describe a range of health impacts during extreme weather events.

There are several strengths to our approach for monitoring ambulance dispatch data. Data are collected using automated daily routines meaning there are no ongoing requirements from the ambulance trusts participating. The surveillance utilizes existing chief complaint codes used in the routine triage of patients and, therefore, does not require trusts to use additional or bespoke codes. There are, however, some limitations. Firstly, the syndromic indicator definitions are non-specific, potentially covering a wide range of disorders, which means that it is sometimes difficult to elucidate the cause of the observed syndromic spikes. In spite of this limitation, we found that NASS heat/cold exposure CPC calls can provide important information to augment and inform situational awareness and alerts for the associated health impact of extreme weather, which can be explored using other data sources. Secondly, due to operational requirements and processes within individual ambulance trusts, there was variation in coding by individual trusts, which means that undertaking the analysis at regional level is challenging. Exploring the regional variation is a key development of the system in the future. Future work would explore regional variation in ambulance call numbers and the impact of regional temperatures, weather patterns, and socio-demographic factors on their distribution. Furthermore, it is important to explore further the utility of other specific chief complaints and how they relate to extreme weather events. The NASS system can be further expanded to cover devolved administrations and make a UK-wide system.

However, NASS is part of a suite of national syndromic systems, and other data streams can inform on regional variations in heat/cold impact.

## 5. Conclusions

Due to climate change, the negative health effects of extreme weather are likely to grow in magnitude and frequency. We believe that as for any other health event, data collection, analysis, and interpretation are key to understanding and mitigating the adverse effects of extreme weather. We show that NASS heat and cold CPC is a highly specific indicator of the negative health impact of extreme weather events in England. The system alarms correspond to those produced the by multi-modal alerting system developed by the Met office. Finally, NASS data correlate with related indictors from other syndromic surveillance systems. These findings show that NASS can be used effectively for surveillance during extreme weather and be triangulated with other data sources to provide a comprehensive quantification of the health impacts associated with extreme weather. NASS data deliver valuable additional information as it is real time, longitudinal and provides data on conditions at the extreme end of the clinical spectrum. The use of NASS data can be further expanded and improved through the investigation to look at how regional temperature changes impact regional call volumes. Localized information would provide information to support local preparedness and response to health and cold weather events. While monitoring for extreme weather events is a key component of ambulance surveillance, NASS has already demonstrated value to a number of other incidents including during the global COVID-19 pandemic. We recommend that NASS is used in combination with other surveillance systems to monitor and quantify the health impact of extreme weather events on the population in England. Integrating NASS into UKHSA’s multi-hazard surveillance program can enhance early warning and situational awareness during extreme weather events

## Figures and Tables

**Figure 1 ijerph-19-03876-f001:**
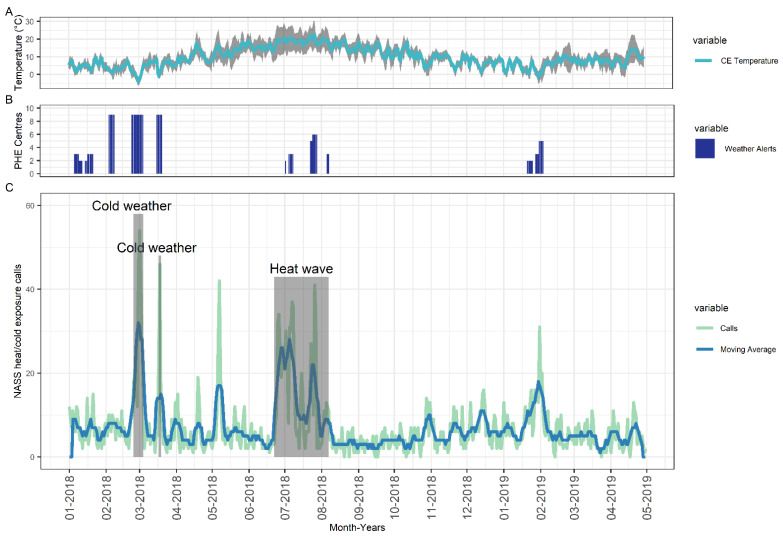
(**A**) mean (blue line), min and max (grey area) central England (CE) temperature (top panel) over time, (**B**) PHE Heat Health and Cold Weather Alerts (middle panel), and (**C**) Number of Heat/cold exposure CPC calls (bottom panel) with extreme weather events identified *a priori* marked on graph in grey boxes.

**Figure 2 ijerph-19-03876-f002:**
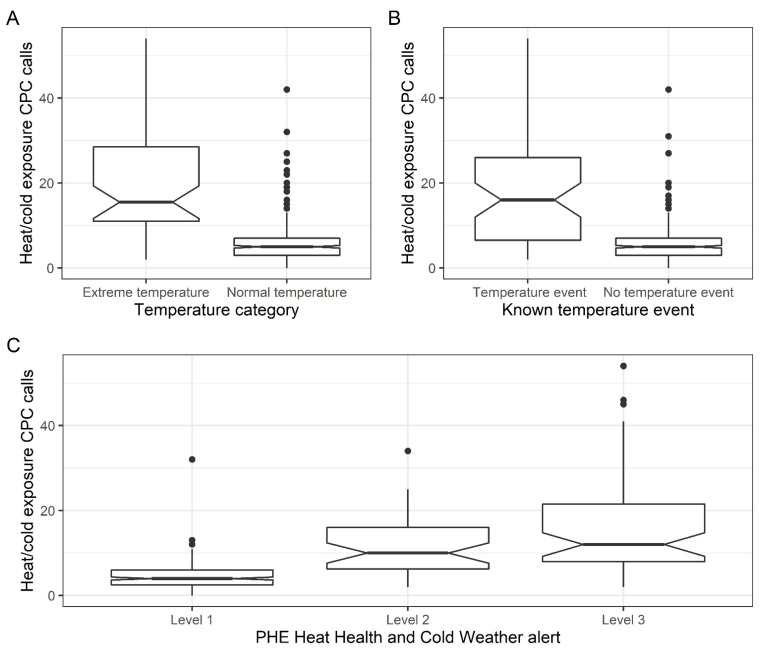
Box-whisker plot describing the (**A** + **B**) number of heat/cold exposure CPC calls observed between extreme and normal temperature days (top left) and KTE, non-temperature event (top right), and (**C**) across different PHE Heat Health and Cold Weather Alert levels (bottom).

**Figure 3 ijerph-19-03876-f003:**
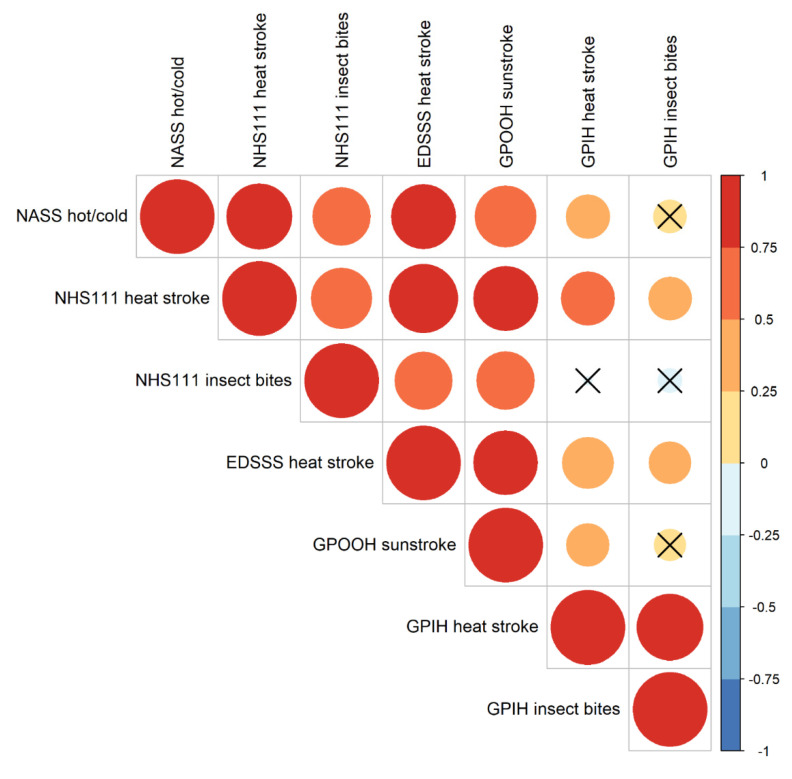
Correlation between NASS heat/cold exposure CPC calls and related syndromic indicators from telehealth (NHS 111), emergency department (EDSSS), GP in hours (GPIH), and GP out of hours’ (GPOOH) systems. Each circle represents a correlation pair (small circles represent a weak and large a strong correlation, respectively), with the color and size dependent on the pairs correlation co-efficient (r). Circles without crosses represent significant correlations and circles with crosses signify non-significant correlation between indicators.

**Table 1 ijerph-19-03876-t001:** Number of a NASS heat/cold exposure CPC statistical alarms by whether at least one region experiencing PHE Heat Health and Cold Weather Alerts.

	Level 1	Percentage	Level 2	Percentage	Level 3	Percentage	Total
No NASS Alarm	245	80%	30	10%	31	10%	306
NASS Alarm	2	5%	12	29%	28	67%	42
Total	247	71%	42	12%	59	17%	348

## Data Availability

The datasets used in this study are not publicly available.

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
