# Peer review of "The Utility of Ambulance Dispatch Call Syndromic Surveillance for Detecting and Assessing the Health Impact of Extreme Weather Events in England"

_ijerph, 2022, doi:10.3390/ijerph19073876_

Round 1
Reviewer 1 Report
This clearly written manuscript describes the utility of the National Ambulance Syndromic Surveillance System (NASS) for identifying health impact day of extreme temperature in England. The methods are straight forward and limitations of the results are included, such as the inability to separate heat from cold complaints.
My primary suggestion would be to consider looking beyond the heat/cold chief presenting complaint syndromes to explore other possible areas of health impacts (e.g., cardiovascular events, mental health crises) to see if this approach could be used to generate new hypotheses for future research. It was also unclear to me whether it was possible to add additional neighborhood or other geospatial factors that might identify communities most at risk where more targeted warnings or public health outreach might be focused to prevent the need for emergency care during extreme temperature events.
Reviewer 2 Report
The structure of the paper can be improved...
The paper contains more than 20 applicable references.
BUT:
1. There are 11 authors, but a short research... It is unusual for a paper with little findings to have 11
authors.
2. You said in the abstract:
"Daily total calls were observed between 01/04/2016-27/03/2019."
but in the text of the paper:
"The study period consisted of 485 days from 01/01/2018 to 30/04/2019."
Please clarify...
+
Please make a bibliographic (with citations) research related to the previous researches/findings in this
field... you can compare the results from previous studies to your findings...
A can say that your related work must be updated...
3. The research findings are predictable, to be expected somehow ...
"The study found that ambulance heat and cold exposure calls had a strong association with severe 186 weather
and temperature events occurring in England. The number of daily calls were generally low, but during extreme
weather events heat and cold ambulance calls increased exponentially."
4. The conclusions have some citations... the conclusions are yours!
Not references to other conclusions from other researches!
5. Related to Figure 3: Correlation between NASS heat/cold exposure CPC calls and related syndromic indicators
179 from NHS 111, GPOOH, GPIH and EDSSS. Each circle represents a correlation pair with the color 180 and size
dependent (small circles represent a weak and large strong correlation respectively) on the 181 pairs R value
and crosses signifying no significant correlation.
But... how did you get to them? Please share all the data and show a mathematical explanation...
- you also said "All analysis were undertaken in R version 4.0.2 (2020-06-22)"
Please explain better...
6. "Discussion" section has citations of other researches! Why?
In this section you must discuss your findings...
7. What are the limitations of your research?
8. What are your future work intentions?
9. In my opinion your paper has a lot of limitations...
please MAJOR review is needed!
Reviewer 3 Report
The paper presents some very straight forward results looking at ambulance data for extreme weather events involving hot and cold temperatures. These events are relatively rare in England but a signal seems to be clear in the ambulance data. This is a fairly obvious finding. It would have been interesting to compare the maximum/minimum temperatures recorded in each ambulance region with the number of heat/cold calls? The Central England Temperature (CET) is an average and does not show the extreme temperatures as Figure 1(a) shows. Please add the minimum and maximum CET temperatures to Figure 1(a).
More regional temperature data needs to be presented to identify thresholds ie at what maximum/minimum temperatures do hot/cold ambulance calls increase? How does this vary across England? Are people in the North more able to cope with cold temperatures and people in the South more able to cope with hot temperatures ie due to acclimatisation?
Other points:
1. lines 19-21 do not make sense?
2.The Abstract has too many acronyms to read easily?
3. line 76 the dates are different to those given in the abstract?
4. line 143 states 16/days the abstract states 15/days?
5. lines 164-167 the sentence does not make sense?
6. In Figure 1(c) what caused the spike in May 2018? it should be possible to look back at the ambulance data to see where the spike came from?
Round 2
Reviewer 2 Report
0. "The structure of the paper can be improved" - This is not my first recommendation, my Comment 1.
My comment 1 is:
1. There are 11 authors, but a short research... It is unusual for a paper with little findings to have 11 authors.
But... in response to the authors...
The authors can be more detailed in Introduction and Results for example... this is what I said...
1. It's still a mismatch.... The author said in the abstract:
Daily total calls were observed between 01/04/2016-30/04/2019
but in the text of the paper:
"The study period consisted of 485 days from 01/01/2018 to 30/04/2019."
2.The authors did not respond enough to my comment:
"Please make a bibliographic (with citations) research related to the previous researches/findings in this field... you can compare the results from previous studies to your findings...I can say that your related work must be updated... "
3.
The authors only deleted the citation [17] from the Conclusion!! and in the coverletter they said:
"Not references to other conclusions from other researches!"
See below:
While monitoring the impact of extreme heat and cold weather events is a key component of ambulance syndromic surveillance, NASS has already been used to respond to a number of other incidents including the global
COVID-19 pandemic [17].
While monitoring the impact of extreme heat and cold weather events is a key component of ambulance syndromic surveillance, NASS has already been used to respond to a number of other incidents including the global
COVID-19 pandemic.
4.The authors did not respond enough to my comment:
But... how did you get to them? Please share all the data and show a mathematical explanation...
- you also said "All analysis were undertaken in R version 4.0.2 (2020-06-22)"
Please explain better...
The authors just said:
The correlation between daily NASS calls and corresponding ‘heatwave’ indicator counts from other UKHSA syndromic surveillance systems (NHS 111 ‘heat and sunstroke’, NHS 111 ‘insect bites’, EDSSS ‘heat/sunstroke’,
GPOOH ‘sunstroke’, GPIH ‘insect bite’ and GPIH ‘heat stroke’) were calculated using Spearman rank correlation coefficients, mid ranks were used for ties and p-values approximated using F distributions. “
The paper has the same structure, the same problems...
In majority of cases the answers are contradictory to what I said...
My decision remain the same.
Reviewer 3 Report
Thank you for your corrections - just one question - line 188 needs correcting.
Round 3
Reviewer 2 Report
I appreciated the authors effort to make corrections. The paper looks better this time. Please contact the editorial team for other information.